# The effects of L-carnitine and fructose in improved Ham's F10 on sperm culture in idiopathic severe asthenospermia within 24h

Dehui Chang[1¤], Fudong Li[1☺], Yindong Kang[1☺], Yixin Yan[1☺], Feiyan Kong[2☺], Wei Jiang[3☺], Dongxing Wang[1☺], Zhigang Cao[1], Liuting Xu[1], Chuang Huang[1], Yafen Kang[1], Xuejun Shang[4‡]*, Bin Zhang [1‡]*

1 Department of Urology, The 940th Hospital of the People's Liberation Army Joint Logistics Support Force, Lanzhou, Gansu, China, 2 Second Department of Surgery, Beijing Fengtai Hospital of Integrated Traditional Chinese and Western Medicine, Beijing, China, 3 Convalescent Section First of Convalescent Zone Second, Air Force Hangzhou Secret Service Rehabilitation Center, Hangzhou, Zhejiang, China, 4 Department of Urology, Nanjing Jinling Hospital: General Hospital of Eastern Theatre Command, Nanjing, Jiangsu, China

☺ These authors contributed equally to this work.
¤ Current address: Department of Medicine, Northwest Minzu University, Lanzhou, Gansu, China
‡ BZ and XS are co-corresponding author on this work.
* 1265243155@qq.com (BZ); shangxj98@163.com (XS)

**Data Availability Statement:** All relevant data are within the manuscript and its Supporting Information files.

## Abstract

To study the effects of L-carnitine and fructose on semen parameters of severe asthenospermia patients by sperm culturing in vitro within 24h. We optimized the energy composition and antioxidant substances of sperm culture medium in vitro (based on Ham's F10 culture medium) by orthogonal test for preparing high quality culture medium. Sperms of 60 patients with idiopathic severe asthenospermia were collected, and cultured in vitro within 24h, by Ham's F10 culture medium added to different concentrations of L-carnitine and fructose and culture temperature, whose effects on sperm motility were observed to determine which is the most appropriate concentration and temperature. For determining the appropriate concentration of L-carnitine and fructose and the suitable culture temperature in Ham's F10 culture medium, the orthogonal experiments were carried out to optimize above three factors, which had great influence on sperm viability, survival rate, deformity rate and DNA fragmentation index (DFI). The final concentration of L-carnitine and fructose was determined in terms of initial tests to assess the effects of different concentrations (4, 8, 12, and 16 mg/ml L-carnitine and 0.125, 0.250, 0.375, and 0.50 mg/ml fructose) on sperm viability and motility in culture. During the operation of processing and culturing sperms in vitro within 24h, orthogonal test showed that sperm viability was better at the final concentration of 8 mg/ml L-carnitine and 0.375 mg/ml fructose in improved Ham's F10 culture medium at 36.5˚C. Idiopathic severe asthenospermia sperm can be effectively improved by the modified Ham's F10 culture medium of the final concentration of 8 mg/ml L-carnitine and 0.375 mg/ml fructose at 36.5˚C within 24h, which has shown better culture effect and is superior to Ham's F10 basic medium.

**Funding:** Supported by the Fundamental Research Funds for the Central Universities and the grant numbers is 31920240072. Supported by the the Health Special Project fund of PLA and the grant numbers is 21BJZ43. Supported by the Special Scientific Research Projects for Family Planning, Prenatal and Postnatal Care of PLA and the grant numbers is 21JSZ13. Supported by the Medical and Health Project fund of the 940th Hospital and the grant numbers is 2021yxky017, 2021. Supported by the Medical and Health Project fund of the 940th Hospital and the grant numbers is 2023YXKY006 and 2023YXKY024, 2023. Supported by the Natural Science Foundation of Gansu Province and the grant numbers is 22JR5RA001, 2022. Supported by the Natural Science Foundation of Gansu Province and the grant numbers is 23JRRA531, 2023. Supported by the Lanzhou science and technology project Bureau and the grant numbers is 2023-2-63, 2023. Dehui Chang and Bin Zhang are the funders of above funds. Dehui Chang contributed to the whole project development. Bin Zhang contributed to carrying out the literature review, the language translation, grammar checking, and manuscript writing.

**Competing interests:** The authors declare that they have no known competing financial interests or personal relationships that could have appeared to influence the work reported in this paper. All authors also declare that they have agreed to participate in this study and agree to publish the obtained data and analysis results of this study.

## Introduction

Human reproductive health has a far-reaching and extensive impact on the world's population, social and economic development, which has attracted widespread attention. In recent years, male reproductive ability has been affected by factors (including environmental degradation, health and safety issues, life pressure, irregular life, and physiological diseases) and has shown a significant downward trend [1–3].

In 2017, "Human Reproductive News" published a research report on the changes in human sperm count. Survey data showed that the global male sperm concentration has dropped by 59.3% in the past 40 years. According to statistics, nearly one-fifth of global couples of childbearing ages have fertility problems, and male factors account for about half of them [4, 5]. Male infertility refers to a couple living together for more than one year, without contraception and having a normal sexual life, and a woman's non-pregnancy caused by the male factor is called male infertility. Male infertility is mainly manifested as azoospermia [6, 7], oligospermia and asthenospermia caused by spermatogenesis disorders. Among them, oligospermia and asthenospermia are the main types of semen abnormalities [8, 9], and are also the main phenotypes of male infertility [10]. Mehra BL et al. summarized the semen quality analysis results of 117979 patients with suspected infertility [11]. The study showed that male infertility was 45%, oligospermia accounted for 22%, asthenospermia accounted for 11%, and azoospermia patients accounted for 12%. Asthenospermia refers to patients with the forward movement spermatozoa is less than 32%, which occurrence is the result of a complex and multi-factorial effect. There are many pathogenic factors, including environmental factors, occupational exposure, chromosomal abnormalities, gene deletions, and infections, Varicocele, iatrogenic diseases, etc [12, 13], which also include changes in sperm morphology and molecular mechanism, abnormalities of mitochondria, abnormalities of dense fibers and fibrous sheaths around the flagella of sperm tails and related to male infertility The abnormal expression of the gene protein and other reasons [14–18].

Asthenospermia is defined as progressive motility < 32% (abstinence for 2–7 days) [19]. At present, in the treatment of asthenospermia, the treatment of secondary male infertility is better, but for idiopathic asthenospermia, understood as a form of asthenospermia with unexplained and unknown etiology. Severe asthenozoospermia refers to patient with 1 ~ 10% (China) [20] or < 20% (Spain) [21] progressive motility. Therefore, idiopathic severe asthenospermia refers to Chinese patient with 1 ~ 10% progressive motility with undiscovered etiology in this study. ART is generally used to help these people get pregnant.

ART is inseparable from the in vitro processing of sperm, and the assisted reproductive culture fluid, as a medium that directly contacts and preserves sperm, is an important factor affecting sperm in vitro culture. Progressive sperm motility has a certain influence on the outcome of ART [22]. High-quality culture medium can not only prolong the survival time of sperm in vitro, but also have nutritional protection, improve sperm motility, and select sperm with good vitality for artificial insemination and in vitro insemination [19], and improve the success rate of assisted pregnancy. The osmotic pressure, pH value and nutrients contained in the culture fluid can affect the vitality and survival time of sperm and the ability of fertilization. In this regard, this experiment studies the effects of L-carnitine, fructose, and appropriate temperature on the in vitro sperm culture of asthenospermia patients, and expects to obtain the best improved sperm culture results in vitro within 24h.

## Methods

### 1. Trial design

**1.1 Determination of appropriate concentrations of L-carnitine and fructose.** *1.1.1 Effects of different concentrations of L-carnitine on sperm culture in vitro within 30min.* **(1) Reagent preparation.** 8 mg/ml L-carnitine working solution: 160 mg of L-carnitine was added to 20 ml of normal saline. After fully mixed and dissolved, L-carnitine solution was filtered through a microporous filter, and then was placed in a sterile bottle for later use. Use the same method to prepare 16, 24 and 32 mg/ml L-carnitine working solution. When equal volume of semen was added, the final concentration was 4, 8, 12 and 16 mg/ml.

**(2) Test grouping.** After complete liquefaction for 30 min in 37°C thermostatic water bath, 60 semen samples were completely randomly selected from patients and equally divided into 6 groups (200 μl in each group), in which the sperm motility was tested as the basic data before culture. The treatment conditions of each group are as follows: Group NS1 and HF1: adding with the same volume (200 μL) of normal saline and Ham's F10 medium. Groups L4, L8, L12, and L16: adding with the same volume (200 μL) of L-carnitine working solution, which the final concentrations was 4, 8, 12, and 16 mg/ml. Normal saline, Ham's F10 medium and L-carnitine working solution were prewarmed in 37°C constant temperature water bathing for 5 min before use. The semen samples of each group were mixed and placed in 37°C constant temperature water bath for 30 min.

*1.1.2 Effects of different concentrations of fructose on sperm culture in vitro.* The reagent preparation, test grouping and processing method was similar to the previous experimental groups: 5 mg of fructose, was added to 20 ml of normal saline (0.250 mg/ml fructose working solution). Use the same method to prepare the different concentration solution. Group NS2 and HF2 were same to 2.1.1.1. Groups F0.125, F0.250, F0.375, and F0.500: adding with the same volume of fructose working solution, which the final concentrations was 0.125, 0.250, 0.375, and 0.50 mg/ml.

### 1.2 Improvement of Ham's F10 base medium

*1.2.1 Orthogonal experiment.* (1) Experimental design: The three factors of L-carnitine, fructose and culture temperature have influence on semen culture in vitro. Therefore, the above three factors were selected for orthogonal optimization of Ham's F10 culture medium. We used three factors and three levels of $L_9$ ($3^4$) orthogonal experimental to formulate the table of optimization factors in sperm culture in vitro (see Table 1) and design $L_9$ ($3^4$) orthogonal experimental table (see Table 2).

(2) Reagent preparation: Preparation of 9 different concentrations of modified Ham's F10 medium according to orthogonal experimental design: 160 mg of L-carnitine and 5 mg fructose were fully dissolved in 20 ml Ham's F10 medium, thoroughly shaken and mixed, filtered through a microwell filter, and placed in a sterile bottle for later use (the final concentrations see Table 2). (3) Test grouping: The selection and treatment of semen samples and working solution were same as the previous experiment and divided into 9 groups (6 cases in each group, see Table 1), in which added the same volume of 9 different concentrations of modified Ham's F10 medium respectively.

### 1.3 Performance test of modified Ham's F10 medium

The selection and treatment of semen samples and working solution were same as the previous experiment and 60 semen samples were equally divided into 3 groups (NS3, HF3 and m-HF,

**Table 1. Orthogonal test.**

| Test No. | factors 1 (L-carnitine) | factors 2 (fructose) | factors 3 (culture temperature) | Results |
|---|---|---|---|---|
| 1 | 4mg/ml | 0.125mg/ml | 36.5°C | 7.51 |
| 2 | 8mg/ml | 0.125mg/ml | 37.0°C | 7.42 |
| 3 | 12mg/ml | 0.125mg/ml | 37.5°C | 7.36 |
| 4 | 4mg/ml | 0.250mg/ml | 37.0°C | 7.63 |
| 5 | 8mg/ml | 0.250mg/ml | 37.5°C | 7.45 |
| 6 | 12mg/ml | 0.250mg/ml | 36.5°C | 7.38 |
| 7 | 4mg/ml | 0.375mg/ml | 37.5°C | 7.31 |
| 8 | 8mg/ml | 0.375mg/ml | 36.5°C | 7.87 |
| 9 | 12mg/ml | 0.375mg/ml | 37.0°C | 7.12 |
| K1 | 22.29 | 22.45 | 22.76 | |
| K2 | 22.46 | 22.74 | 22.17 | |
| K3 | 22.3 | 21.86 | 22.12 | |
| k1 | 7.43 | 7.48 | 7.59 | |
| k2 | 7.49 | 7.58 | 7.37 | |
| k3 | 7.43 | 7.29 | 7.39 | |
| R | 0.06 | 0.29 | 0.21 | |

20 cases in each group), in which added the same volume of normal saline, Ham's F10 medium and modified Ham's F10 medium respectively.

## 2. Ethics statement, participants and semen specimen collection

Ethics approval by the Human Ethics Committee of the 940 Hospital (IRB approval No. 2016KYLL086). And the use of this experimental semen specimens and the experimental protocol were agreed. The clinical trial number: 2016CLT-U002.We informed each participant the details of the study: We need to collect some semen samples from you, and only use the remaining semen samples from the routine semen test. The sample is only used in this study and will not affect routine treatment or diagnosis. Written informed consent forms were obtained from each participants included in this study.

The semen specimens were collected from 60 patients with idiopathic severe asthenozoospermia (the criteria was that forward motility of sperm is 1 ~ 10% and with unknown etiology), aged 24–35 (26.5±4.31) years, participated in this study at the Urology Center of the 940th Hospital, from September 2017 to August 2018. Exclusion criteria were as follows: (1) abnormal chromosome examination; (2) cryptorchidism or testicular dysplasia; (3) acute urogenital

**Table 2. Variance analysis.**

| Source of variation deviation | n | Sum of squared deviations | variance | F value | Fa | Significant level |
|---|---|---|---|---|---|---|
| factors 1 (L-carnitine) | 6 | 0.006 | 0.034 | 5.143 | F0.3(2,4) = 2.374 | P<0.01 |
| factors 2 (fructose) | 6 | 0.134 | 0.067 | 22.293 | F0.05(2,4) = 6.944 | P<0.05 |
| factors 3 (culture temperature) | 6 | 0.084 | 0.042 | 14.042 | F0.1(2,4) = 4.325 | P<0.05 |
| Error e | | 0.006 | 0.003 | | F0.25(2,4) = 2.000 | |
| Fixed error e | | 0.012 | 0.003 | | | |
| Sum | | 0.230 | | | | |

tract infection; (4) moderate or severe varicocele; (5) taking drugs that affected reproductive function or receiving treatment (such as hormone drugs, some proprietary Chinese medicines, radiotherapy and chemotherapy, etc.) within 3 months; (6) abnormal sex hormone levels (testosterone, follicle-stimulating hormone, luteinizing hormone, estrogen diol, and prolactin); (7) semen liquefaction time > 1 hour; (8) semen volume of per ejaculation < 1.5 ml.

All specimens were collected 2 to 7 days following sexual abstinence, which were processed and analyzed according to WHO laboratory manual for the examination and processing of human semen (5th Edition).

NOTE: Clinical researchers were divided into two groups: one group was responsible for enrolled participants, collected semen samples, and generated the random allocation sequence. The other group was responsible for statistical analysis of the experimental data. Double blindness was performed between the two groups to determine the reliability of experimental results. Since the subjects of this study were semen samples from these patients with idiopathic severe asthenozoospermia, whose semen quality was significantly lower than that normal males, the results of different concentration of L-carnitine and Fructose were only compared with results of normal saline and Ham's F10 medium before and after culture.

## 3. Test equipment and reagents

(1) 5 μL, 100 μL pipette (Shanghai Dalong Medical Equipment Co., Ltd.), (2) 200 μL pipette (Eppendorf, Germany), (3) WLJY-9000 sperm quality analyzer (Beijing Weili New Century Technology Development Co., Ltd.), (4) Microporous filter (Shanghai Shupei Experimental Equipment Co., Ltd.), (5) Constant temperature water bath (Shanghai Yuejin Medical Equipment Co., Ltd.), (6) L-carnitine (Santacmz, USA), (7) Fructose (Dongguan Warisi Chemical Co., Ltd.), (8) Ham's F10 medium (Gibco, USA), (9) Sperm morphology rapid staining solution (Diff-Quik method, Shanghai Soleibo Biotechnology Co., Ltd.), (10) Sperm in vivo staining solution (eosin method, Shanghai Soleibo Biotechnology Co., Ltd.), (11) Wright's staining solution (sperm chromatin diffusion method, Shanxi Kangbao Biotechnology Co., Ltd.).

## 4. Observation indicators

**4.1 Indicators in 2.1.1 tests.** The sperm motility of each group was calculated again after culture in 37˚C constant temperature water bath for 30 min, compared with the basic data in group NS1 collected before and HF1 after culture.

**4.2 Indicators in 2.1.2 test.** We calculated the percentage of forward motile spermatozoa, percentage of non-forward motile spermatozoa, malformation rate, survival rate and sperm DNA fragmentation index after 30 min, 1h, 12h, and 24h of culture. In particular, it should be noted that in vitro culture temperature was 36.5˚C.

## 5. Statistical analysis

SPSS 22.0 statistical software were used for the data statistics. Quantitative data conforming to normal distribution and homogeneity of variance were expressed as ($\bar{x} \pm s$). Paired sample t test was used for inter-group measurement data. $P < 0.05$ was considered statistically significant.

## Results

## 1. Determination of appropriate concentrations of L-carnitine in vitro within 30min

Grouping see 2.1.1.1. Compared with the basic data before culture and results of culture in Ham's F10 medium, there was decreased in the percentage of forward motility and non-

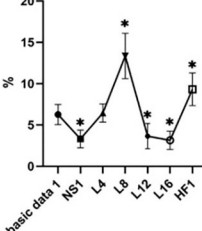

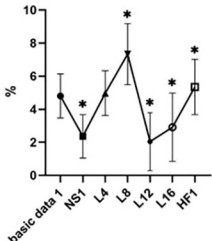

**a1: Percentage of forward motility of sperm (%)**

**a2: Percentage of non-forward motility of sperm (%)**

**Fig 1. Changes of sperm motility of groups in different concentrations of L-carnitine culture ($\bar{x} \pm s$).** Note: *: VS Basic data1, P<0.05.

forward motility of sperm in groups NS1, L12 and L16 after culture (the difference was significant, P<0.05). But there was no significant difference among groups NS1, L12, and L16 after culture (P>0.05). There was increased percentage in groups L4, L8, and HF1 after culture. But compared with the basic data, the difference was not significant in group L4 (P>0.05) and the difference was significant in group L8 and HF1 (P<0.05) after culture. The test results were shown in Fig 1.

## 2. Determination of appropriate concentrations of fructose in vitro within 30min

Grouping see 2.1.2.1. Compared with the basic data before culture and results of culture in Ham's F10 medium, there was decreased in the percentage of forward motility and non-forward motility of sperm in group NS2 after culture (P<0.05). But there was no significant difference among group F0.125, and F0.250, and HF after culture (P>0.05).

There was an increased percentage in group F0.375 and F0.500 after culture (P<0.05). The increasing degree in group F0.375 was greater than F0.500 in the percentage of forward motility of sperm (P<0.05). But there was no significant difference between group F0.375 and F0.500 in the percentage of non-forward motility of sperm after culture (P>0.05). The test results were shown in Fig 2.

## 3. Improvement of Ham's F10 base medium

Grouping see 2.1.3. After 30 min culture, the order of variance values of three factors was L-carnitine > culture temperature > fructose, according to the results of variance analysis in

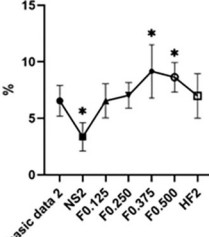

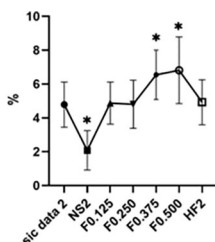

**b1: Percentage of forward motility of sperm (%)**

**b2: Percentage of non-forward motility of sperm (%)**

**Fig 2. Changes of sperm motility in different concentrations of fructose culture groups ($\bar{x} \pm s$).** Note: *: VS Basic data2, P<0.05.

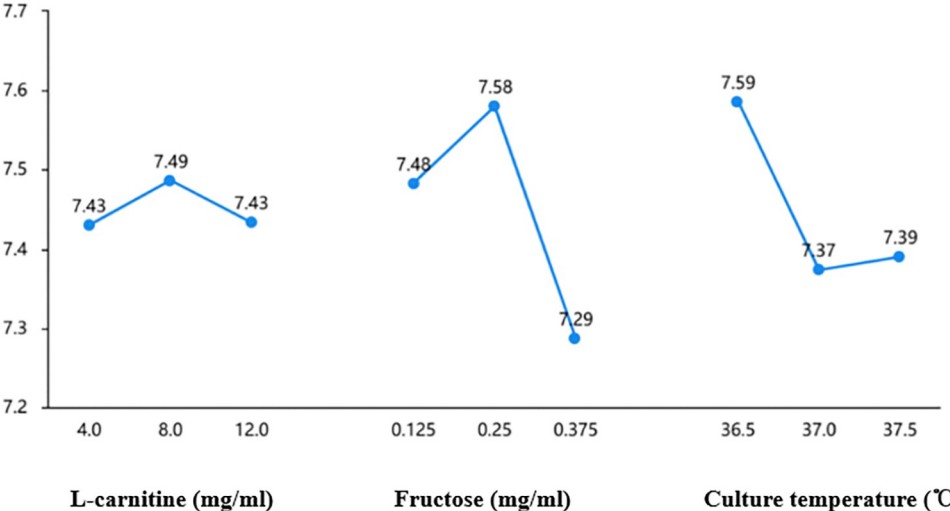

**Fig 3. Effect diagram of orthogonal test.**

Table 2, which indicated that L-carnitine had the greatest influence on the test results, while fructose had the least influence on the test results. In addition, group 8 (the final concentrations was 8 mg/ml L-carnitine and 0. 375 mg/ml fructose) and C1 (36.5˚C culture temperature) could obtain the highest percentage of forward motioning sperm. The results of orthogonal test were shown in Table 1. The results of Analysis of Variance were shown in Table 2. And the effect curve was shown in Fig 3.

## 4. Performance test of modified Ham's F10 medium

**4.1 Changes of semen parameters in each groups after 30 min in each group.** Compared with the basic data3 before culture, there was increased in the percentage of forward motility and non-forward motility of sperm in groups HF3 and m-HF3 after culture (the difference was significant, $P<0.05$). There was increased in Sperm DNA fragmentation rate in group m-HF (the difference was significant, $P<0.05$). The test results are shown in Fig 4.

**4.2 Changes of semen parameters in groups after 1 h in each group.** Compared with the basic data3 before culture, there was increased in the percentage of forward motility and non-forward motility of sperm in groups HF3 and m-HF after culture (the difference was significant, $P<0.05$). There was increased in Sperm DNA fragmentation rate in group m-HF (the difference was significant, $P<0.05$). The test results are shown in Fig 5.

**4.3 Changes of semen parameters in groups after 12 h in each group.** Compared with the basic data3 before culture, there was decreased in the percentage of forward motility, non-forward motility and survival rate of sperm in groups NS3, and HF3. There was increased in deformity rate and sperm DNA fragmentation rate in groups NS3 after culture (the difference was significant, $P<0.05$). The test results are shown in Fig 6.

**4.4 Changes of semen parameters between groups after 24 h in each group.** Compared with the basic data3 before culture, except for the percentage of non-forward motility of sperm was no significant difference in Group m-HF ($P>0.05$), other results were all changed in group NS3, HF3, and m-HF (the difference was significant, $P<0.05$). The test results are shown in Fig 7.

**4.5 The Figures of Semen DNA fragmentation index staining of the groups within 24h.** Compared with the basic data3 before culture, Semen DNA fragmentation index was no

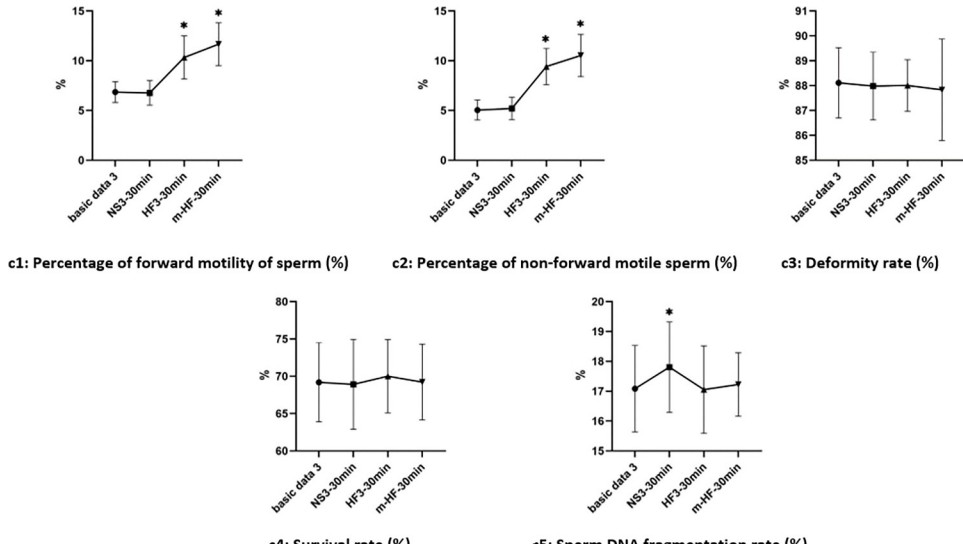

**Fig 4. Changes of semen parameters in groups after 30 min culture.** Note: *: VS Basic data3, P<0.05.

significant difference in Group m-HF-30min, m-HF-1h, and m-HF-12h (P>0.05), other results were all increased in group NS3, HF3, and m-HF-24h (the difference was significant, P<0.05). The test results are shown in Fig 8.

## Discussion

Male infertility has increased significantly in recent years [23], which the common causes are abnormal and continuous decline of sperm quality. Medical researchers have made remarkable achievements on male infertility from basic laboratory research, prevention, to clinical treatment and other aspects of continuous exploration [24–27]. For those with abnormal sperm quality caused by varicocele, surgery and conservative treatment with medicine can be used to improve sperm quality to achieve natural conception [28–32]. The various anti-infective and

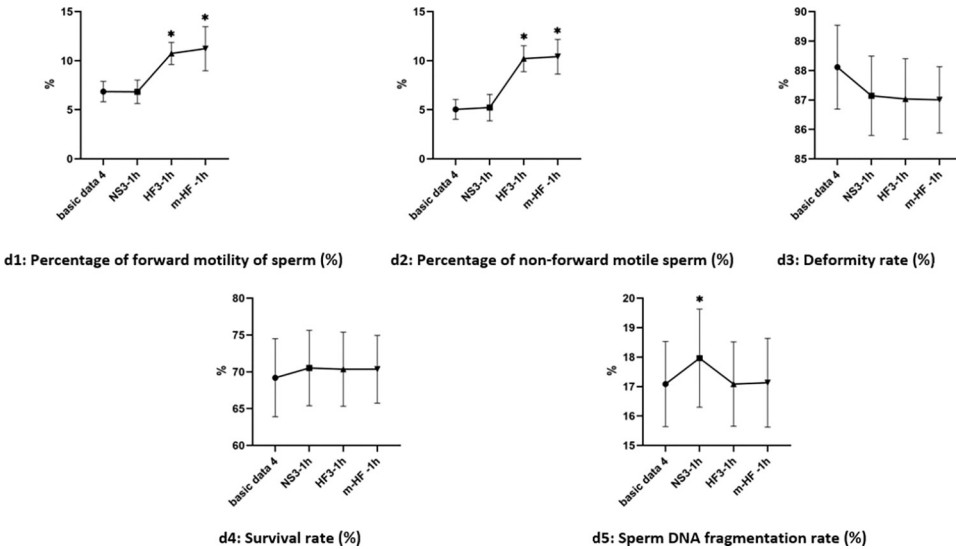

**Fig 5. Changes of semen parameters in groups after 1 h culture.** Note: *: VS Basic data3, P<0.05.

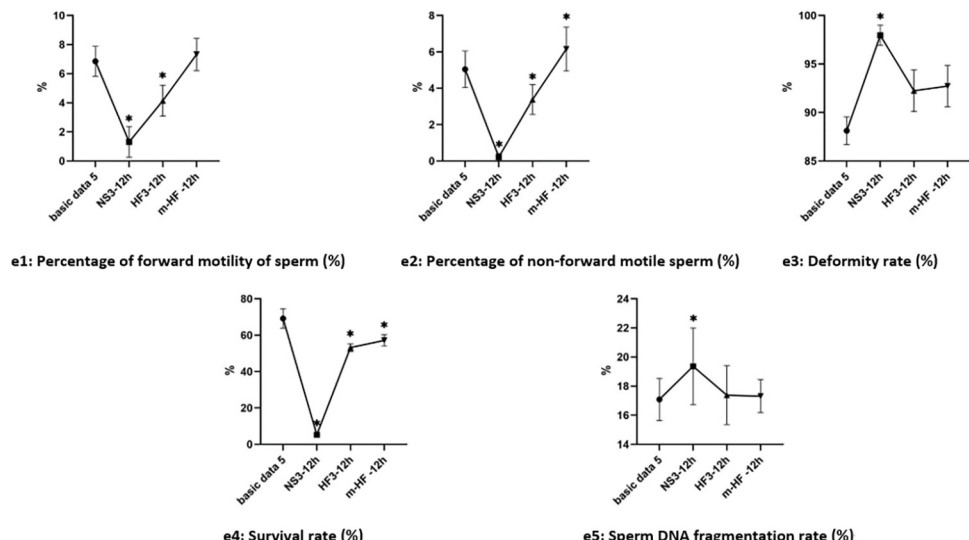

**Fig 6. Changes of semen parameters in groups after 12 h culture.** Note: *: Basic data3, P<0.05.

antiviral therapies have been studied so far in order to ameliorate reproductive system viral infection in order to combat the harmful consequences leading to male infertility [33, 34]. However, there are still a part of asthenospermia with unclear etiology, who have accepted hormonal therapy or empirical treatment. But the effect is uncertain [35]. The application of ART has brought hope to such patients with idiopathic severe asthenospermia (1 ~ 10% progressive motility), and become the most effective treatment for infertility at present [36]. Both intra-uterine insemination and intrauterine sperm injection can achieve a high success rate [37]. The quality of sperm is the key to the success of ART [38]. Therefore, how to improve the quality of sperm through ART is a problem faced by medical researchers.

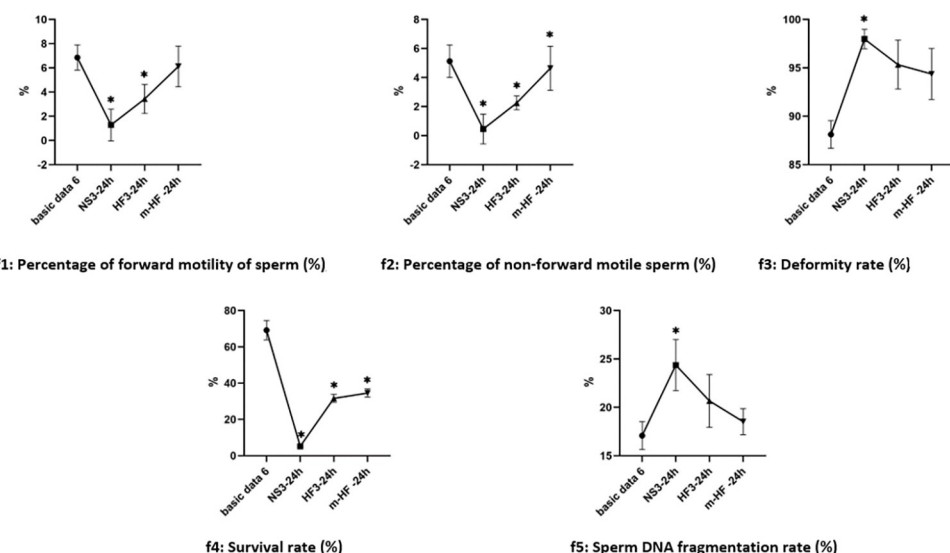

**Fig 7. Changes of semen parameters between groups after 24h culture.** Note: *: VS Basic data3, P<0.05.

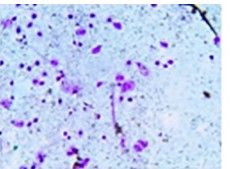

**Basic Data3**

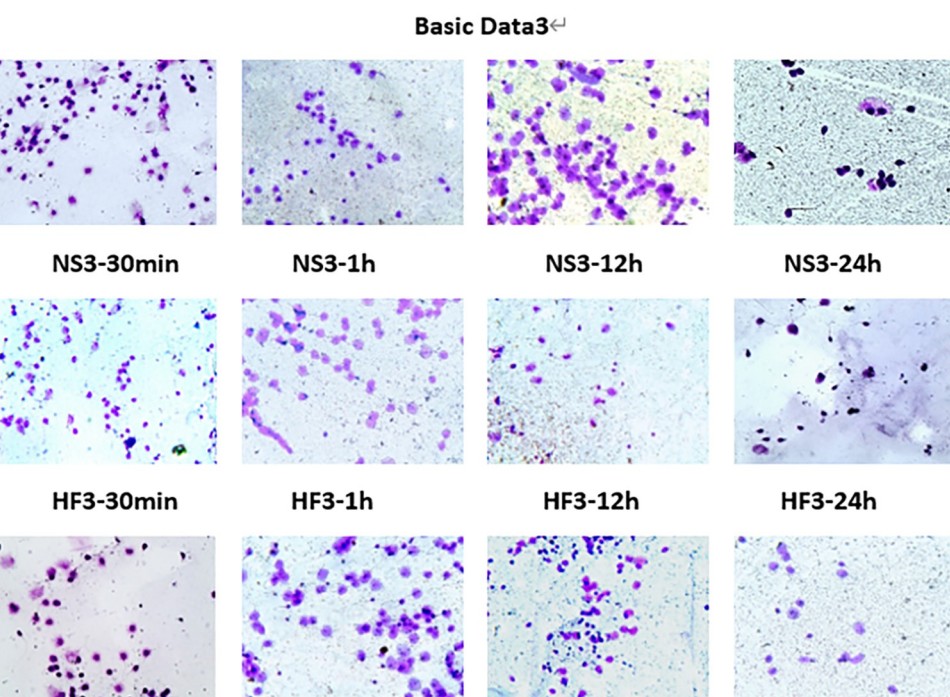

**Fig 8. Semen DNA fragmentation index staining of the groups basic data3, NS3, HF3, and m-HF after 30min, 1h, 12h, and 24h culture (light microscope X400).** Note: Semen DNA fragmentation index reflect the degree of sperm DNA integrity and the ultrastructural damage of sperm. The more damaged and fragmented sperms appear, the higher the fragmentation rate increased, which results in poor integrity and less halos on the figures. On the contrary, halo is more with normal sperms. This clinical trial was only to test the efficacy of modified Ham's F10 medium on semen culture in vitro for patients with idiopathic severe asthenospermia. It has not entered the clinical trial stage for ART assisted pregnancy treatment. Therefore, there was no clinical outcome and harms for the patients involved in the study.

Currently, methods to optimize sperm have their own advantages and disadvantages, include sperm washing method, upstream method, density gradient method, and glass fiber filtration method, etc., but each [39–41]. Sperm culture in vitro provides a new idea for sperm optimization, which can solve the problem of sperm storage and improving motility in vitro [42–44]. Spermatozoa culture can be obtained from spermatogonial stem cells, testicular spermatogenic cells and direct culture of spermatozoa cells [45], in which the former two only stays in the laboratory stage and the last gets mature application at present [46]. High quality sperm can be obtained through spermatozoa culture in vitro, which can not only improve the success rate of ART for pregnancy, but also help to improve the method of ART for pregnancy [47], and reduce medical costs and risks. There are many kinds of spermatozoa culture medium on the market can be chose. The traditional classical culture media include Ham's F10, Earle's [48], and Tyrode's [49]. Some scholars [48–50] have conducted relevant studies on the performance of these three kinds of culture media. As a classic culture medium, Ham's F10

culture medium has a good effect on cell culture and sperm culture. Hosseini A et al. [50] improved the composition of Ham's F10 (adding 20% HAS) and achieved good results. L-carnitine and fructose, which have significantly improved sperm motility and other parameters in vitro, were verified. It was found that L-carnitine had a good biological effect in the process of spermatozoa culture in vitro [51], which is why L-carnitine is widely used in the field of reproductive medicine, especially in the treatment of asthenospermia [52–56]. Studies have shown that the culture medium containing fructose is superior to the culture medium containing glucose, which the possible reason appears to be related to variations in the sensitivity of hexokinase activity [57].

Based on the above research theory and foundation, We designed and conducted this study: adding L-carnitine and fructose to Ham's F10 culture medium, which are the current research hotspot. According to the osmotic pressure and PH value of sperm culture medium reported in the WHO (World Health Organization) laboratory manual for the examination and processing of human semen (the Fifth Edition, February 2011), the osmotic pressure was adjusted to 290mOsmol/kg. $H_2O$ and PH value was 7.6, and the culture temperature was also studied. Through orthogonal experiment design, the above factors affecting sperm culture were optimized and combined to obtain a better combination scheme of the above factors. And the improved Ham's F10 culture medium was compared with the original Ham's F10 culture medium after sperm culture to find the advantages and disadvantages of the improved performance, which could provide the short-term spermosata culture method in vitro within 24h for the idiopathic severe asthenospermia patients accepted ART pregnancy treatment in the future.

In our experiments, spermatozoa were cultured in vitro with different concentrations of L-carnitine, and we found that it was increased in the percentage of forward motility and non-forward motility of sperm in group C1 (8mg/ml L-carnitine) ($P<0.05$), especially the percentage of forward motility spermatozoa increased significantly, compared with the basic data before culture and other 4 groups. It was also found that the high concentration of L-carnitine at 12 mg/ml and 16 mg/ml could not promote the change of sperm motility, but inhibit the sperm motility, and reduce the percentage of sperm forward motility and non-forward motility.

As an important energy material of sperm, fructose plays an important role in sperm motility and fertilization. But it has not been reported about spermatozoa culture in vitro with a single fructose medium [57]. We found that the appropriate concentration of fructose could also improve the motility of sperm, from patients with idiopathic severe asthenospermia, cultured in vitro. That is, when the fructose concentration was 0.375mg/ml, the effect was more obvious. But with the further increase to 0.5 mg/ml fructose, sperm motility did not improve as good as 0.375mg/ml.

The culture temperature condition was one of the influence factors on the spermatozoa culture process. Therefore, orthogonal experiments were carried out to optimize the three factors of L-carnitine [58, 59], fructose and culture temperature, which were shown in Table 1. According to the results of variance analysis shown in Table 2, L-carnitine has the greatest influence on the percentage of sperm forward movement in the test results. In addition, the results have shown that the percentage of forward motile sperm could be significantly increased by improved Ham's F10 medium working solution (8 mg/ml L-carnitine and 0.375 mg/ml fructose) in 36.5°C culture temperature. This improved culture medium can be used for the treatment and spermatozoa culture from patients with idiopathic severe asthenospermia in vitro, which has better performance than the original Ham's F10 culture medium.

In order to verify the performance of modified Ham's F10 medium working solution in the short-term spermatozoa culture and storage in the process of ART for pregnancy treatment,

we ran the validation experiment again. Semen parameters were calculated after 30 min, 1h, 12h, and 24h culture in vitro with modified Ham's F10 medium working solution, including the percentage of forward motile spermatozoa, percentage of non-forward motile spermatozoa, malformation rate, survival rate and sperm DNA fragmentation index. The results has shown that modified Ham's F10 medium working solution has obvious advantages in promote the motility of forward sperm within 1 hour. Even after 24 hours, the modified working solution can also sustain the motility of forward sperm, which is no significant difference from the baseline data. The data of the modified working solution after 24h culture in vitro are also close to the baseline data (including sperm deformity, survival rate and DNA fragments). These results indicated that improved Ham's F10 medium working solution could have stable performance and ability to sustain the motility and integrity of sperm cell membranes and DNA within 24 hours.

ART is generally used for the treatment of idiopathic severe asthenospermia of unknown etiology [19, 20, 60]. It is very important factor that the sperm culture medium directly contacts and preserves sperm in the in vitro operation of ART. Our modified Ham's F10 culture medium can prolong the survival time of sperm and improve the sperm motility in vitro, which is more beneficial to screen out better-quality sperm in the subsequent process of intracytoplasmic sperm injection. But the success rate of assisted pregnancy is still needed to confirm by the clinical trials.

## Main conclusions and research prospect

ART is suitable for patients with idiopathic severe asthenospermia, in Which the high-quality culture medium is the key for improve sperm motility.

In this study, Ham's F10 culture medium supplemented with 8mg/ml L-carnitine and 0.375 mg/ml fructose can achieve a better culture effect at 36.5˚C within 24h, which could inhibit the increase of sperm malformation rate and sperm DNA fragmentation index, and effectively improve sperm motility and survival rate. This improved culture medium is suitable for short-term sperm processing of patients with idiopathic severe asthenospermia before male gametes combine with female gametes.

However, the stability and function of this modified culture medium still need to be further verified by large samples, so as to apply to clinical practice and achieve good clinical application.

## Supporting information

**S1 File.**
(DOCX)

**S2 File.**
(DOCX)

**S3 File.**
(DOCX)

**S4 File.**
(DOCX)

**S5 File.**
(DOCX)

**S6 File.**
(DOCX)

## Acknowledgments

We gratefully acknowledge the valuable cooperation of Rong Wang (Clinical Pharmacology Experimental Center, the 940th Hospital of PLA Joint Logistic Support Force) in helping to design orthogonal test of this study.

## Author Contributions

**Conceptualization:** Dehui Chang, Xuejun Shang.

**Funding acquisition:** Dehui Chang, Bin Zhang.

**Investigation:** Yixin Yan, Dongxing Wang, Zhigang Cao, Liuting Xu, Chuang Huang, Yafen Kang.

**Methodology:** Yixin Yan, Xuejun Shang.

**Writing – original draft:** Bin Zhang.

**Writing – review & editing:** Fudong Li, Yindong Kang, Feiyan Kong, Wei Jiang, Bin Zhang.

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
