## [Decision Letter · Decision Letter 0]

9 Aug 2024

PONE-D-24-23439The effects of L-carnitine and fructose in improved Ham's F10 on sperm culture in idiopathic severe asthenospermia within 24hPLOS ONE

Dear Dr. Zhang,

Thank you for submitting your manuscript to PLOS ONE. After careful consideration, we feel that it has merit but does not fully meet PLOS ONE’s publication criteria as it currently stands. Therefore, we invite you to submit a revised version of

the manuscript that addresses the points raised during the review process. Especially the points from Reviewer #2 that the author should introduce the definition, occurrence, ART outcome, ways to improve ART outcome in men with idiopathic severe asthenospermia, as well as how to recruit these patients.

We look forward to receiving your revised manuscript.

Kind regards,

Xuejiang Guo, Ph.D.

Academic Editor

PLOS ONE

Journal Requirements:

3.Thank you for stating the following financial disclosure: "Supported by the Fundamental Research Funds for the Central Universities and the grant numbers is 31920240072.

Supported by the the Health Special Project fund of PLA and the grant numbers is 21BJZ43.

Supported by the Special Scientific Research Projects for Family Planning, Prenatal and Postnatal Care of PLA and the grant numbers is 21JSZ13.

Supported by the Medical and Health Project fund of the 940th Hospital and the grant numbers is 2021yxky017, 2021.

Supported by the Medical and Health Project fund of the 940th Hospital and the grant numbers is 2023YXKY006 and 2023YXKY024, 2023.

Supported by the Natural Science Foundation of Gansu Province and the grant numbers is 22JR5RA001, 2022. 

Supported by the Natural Science Foundation of Gansu Province and the grant numbers is 23JRRA531, 2023.

Supported by the Lanzhou science and technology project Bureau and the grant numbers is 2023-2-63, 2023."

4. We note that your Data Availability Statement is currently as follows: "All relevant data are within the manuscript and its Supporting Information files."

Reviewers' comments:

Reviewer's Responses to Questions

**Comments to the Author**

1. Is the manuscript technically sound, and do the data support the conclusions?

Reviewer #1: Yes

Reviewer #2: Yes

2. Has the statistical analysis been performed appropriately and rigorously? 

Reviewer #1: Yes

Reviewer #2: Yes

3. Have the authors made all data underlying the findings in their manuscript fully available?

Reviewer #1: Yes

Reviewer #2: Yes

4. Is the manuscript presented in an intelligible fashion and written in standard English?

Reviewer #1: Yes

Reviewer #2: No

5. Review Comments to the Author

Reviewer #1: In this study, the semen of patients with severe asthenospermia was collected and studied. Based on Ham’s F10 medium, different concentrations of L-carnitine and fructose were added into, and different culture temperatures were used to study the effects of the above three factors on the sperm motility, survival rate, deformity rate and DNA fragmentation index of sperm cultured in vitro for 24 hours. The meanings of this study are to prolong the survival time of sperm in vitro, improve the quality and motility of sperm, and improve the success rate of clinical assisted reproductive technology. This study has certain clinical significance.

Through orthogonal test, the results of this study found that when 8mg/mL L-carnitine, 0.375mg/mL fructose were added into Ham 's F10 basic medium at the temperature of 36.5 °C, the semen quality of patients with severe asthenospermia could be effectively improved, which was better than that treat with Ham’s F10 basic medium. However, there are several defects that need to be fixed before the article can be accepted:

1. The writing is not standardized, such as the mixed use of mg/mL and mg/ml.

2.When the table crosses the page, the table needs to be adjusted to one page or the continuation table is used.

3.In the study, there were many groups but the abbreviations of the group names were not direct enough and could not be understood at a glance. Such as the saline group could be abbreviated as NS, the Ham’s F10 basic medium group could be abbreviated as HF, the different concentrations of L-carnitine group could be abbreviated as L+ numbers, and the different concentrations of fructose group could be abbreviated as F+ numbers.

4.There are writing errors in the text that need to be corrected. For example, in 3.4.5, there was no significant difference, but marked with p<0.05.

Reviewer #2: Idiopathic severe asthenospermia is a common condition for male infertility, and with relatively bad result even with ART, thus the research on improving ART outcome on these patients is desired. This research found L-carnitine and fructose can improve sperm motility, which may be used to improve ART outcome.

1. The authors should focus on the idiopathic severe azoospermia. In the introduction and discussion, as well as in the method part, the author should introduce the definition, occurrence, ART outcome, ways to improve ART outcome in men with idiopathic severe azoospermia, as well as how to recruit these patients.

2. Some figures should put into the supplementary data, only keeping important results in the main document.

3. Revision of language is suggested.

6. PLOS authors have the option to publish the peer review history of their article (what does this mean?). If published, this will include your full peer review and any attached files.

Reviewer #1: No

Reviewer #2: No

---

## [Author Response · Author response to Decision Letter 0]

5 Sep 2024

Author’s Response to Reviewers‘ Comments

Ms. No.: PONE-D-24-23439

Title: The effects of L-carnitine and fructose in improved Ham's F10 on sperm culture in idiopathic severe asthenospermia within 24h

Journal: PLOS ONE

Editor and Reviewer comments:

Especially the points from Reviewer #2 that the author should introduce the definition, occurrence, ART outcome, ways to improve ART outcome in men with idiopathic severe asthenospermia, as well as how to recruit these patients.

Reply:

According to WHO laboratory manual for the examination and processing of human semen (5th Edition), asthenospermia is defined as progressive motility < 32% (abstinence for 2-7 days). Idiopathic asthenospermia is understood as a form of asthenospermia with unexplained and unknown etiology. However, the definition of severe asthenozoospermia is different in different race. Severe asthenozoospermia refers to patient with 1 ~ 10% (China) or < 20% (Spain) progressive motility. Therefore, idiopathic severe asthenospermia refers to Chinese patient with 1 ~ 10% progressive motility with undiscovered etiology in this study. Although conventional in vitro fertilization (IVF) or intracytoplasmic sperm injection (ICSI) can lead to successful pregnancy in couples with only male infertility, progressive motility has a certain influence on the outcome of ART. Therefore, using sperm culture medium to improve sperm motility as much as possible is helpful to improve ART outcome. The above detailed explanations are supported by updated literature. 

Ethics approval by the Human Ethics Committee of the 940 Hospital (IRB approval No. 2016KYLL086). And the use of this experimental semen specimens and the experimental protocol were agreed. The clinical trial number: 2016CLT-U002.We informed each participant the details of the study: We need to collect some semen amples from you, and only use the remaining semen samples from the routine semen test. The sample is only used in this study and will not affect routine treatment or diagnosis. We obtained written informed consent forms from each participants included in this study.

We recruited 60 patients with idiopathic severe asthenozoospermia (the criteria was that forward motility of sperm is 1 ~ 10% and with unknown etiology), aged 24-35 (26.5±4.31) years, participated in this study at the Urology Center of the 940th Hospital, from September 2017 to August 2018. We informed each participant the details of the study: We need to collect some semen samples from you, and only use the remaining semen samples from the routine semen test. The sample is only used in this study and will not affect routine treatment or diagnosis. Informed consent forms were obtained from the patients included in this study.

Reply to Journal Requirements:

1. We update the manuscript to meet PLOS ONE's style requirements, including those for file naming. 

2. We provide additional details regarding participant consent, which is stated in the ethics statement in the Methods and online submission information. Informed consent forms were obtained from the patients included in this study. We upload the Informed consent for clinical participant. 

3. Dehui Chang and Bin Zhang are the funders of above funds. Dehui Chang contributed to the whole project development. Bin Zhang contributed to carrying out the literature review, the language translation, grammar checking, and manuscript writing.

4. We confirm at this time the submission contains all raw data required to replicate the results of this study. We submit and upload the following data:

- Some minor data errors have been corrected, and the icons has been corrected..

5. By using PACE digital diagnostic tool, We transformed and uploaded the figure files.

Reply to Reviewer #1:

1. Thank you for your review. We standarded the writing and corrected mg/mL to mg/ml.

2. When the table spreads, we have adjusted the table to one page or used a continuation table.

3. According to your suggestion, we rewrote the abbreviations of the group names as follows:

A1: NS, B1: L4, C1: L8, D1: L12, E1: L16, F1:HF.

A2: NS, B2: F0.125, C2: F0.250, D2: F0.375, E2: F0.500, F2:HF.

A3:NS (NS--30min, NS3-1h, NS3-12h, and NS3-24h), B3:HF (HF3-30min, HF3-1h, HF3-12h, and HF3-24h), C3: m-HF (m-HF -30min, m-HF -1h, m-HF -12h, and m-HF -24h).

4. We corrected all writing errors in the text.

Reply to Reviewer #2:

1. In the introduction, we have introduced the definition, occurrence, ART outcome and methods to improve ART outcome in patients with idiopathic azoospermia and idiopathic severe asthenospermia in detail. In the discussion, we emphasized the significance of this study for improving sperm motility in patients with idiopathic asthenospermia in vitro.

2. Because the contents of table 1 and table 2 are duplicated, we deleted table 1 and kept table 2 and 3. We put some figures into the supplementary data, and kept important results in the main document.

3. This paper has been significantly revises. Please refer to a website to help modify the language of the article. The editor’s native language is English. Thank you!

Thank you for your help and support!

Your sincerely yours.

Bin Zhang

2024-09-05

---

## [Decision Letter · Decision Letter 1]

25 Oct 2024

PONE-D-24-23439R1The effects of L-carnitine and fructose in improved Ham's F10 on sperm culture in idiopathic severe asthenospermia within 24hPLOS ONE

Dear Dr. Zhang,

Thank you for submitting your manuscript to PLOS ONE. After careful consideration, we feel that it has merit but does not fully meet PLOS ONE’s publication criteria as it currently stands. Therefore, we invite you to submit a revised version of the manuscript that addresses the points raised during the review process.

We look forward to receiving your revised manuscript.

Kind regards,

Xuejiang Guo, Ph.D.

Academic Editor

PLOS ONE

Journal Requirements:

Reviewers' comments:

Reviewer's Responses to Questions

**Comments to the Author**

1. If the authors have adequately addressed your comments raised in a previous round of review and you feel that this manuscript is now acceptable for publication, you may indicate that here to bypass the “Comments to the Author” section, enter your conflict of interest statement in the “Confidential to Editor” section, and submit your "Accept" recommendation.

Reviewer #2: All comments have been addressed

2. Is the manuscript technically sound, and do the data support the conclusions?

Reviewer #2: Yes

3. Has the statistical analysis been performed appropriately and rigorously? 

Reviewer #2: Yes

4. Have the authors made all data underlying the findings in their manuscript fully available?

Reviewer #2: Yes

5. Is the manuscript presented in an intelligible fashion and written in standard English?

Reviewer #2: Yes

6. Review Comments to the Author

Reviewer #2: The authors have addressed all my comments. However, they added a paragraph in the INTRODUCTION part for the azoospermia. Deletion of this paragraph is suggested since this work is not related to the azoospermia at all.

7. PLOS authors have the option to publish the peer review history of their article (what does this mean?). If published, this will include your full peer review and any attached files.

Reviewer #2: No

---

## [Author Response · Author response to Decision Letter 1]

16 Nov 2024

Reply to Journal Requirements:

1. At present, we have no laboratory protocols, and we will strengthen our work in this area. Thank you for your proposal.

2. The references have been revised and adjusted. The DOI and PMID code is supplemented in the reference [6], [19], [20], [21], and [22]. The original reference [18] was adjusted to reference [7], and references [7-17] was adjusted to reference [8-18].

3. We reviewed the reference list and ensure that it is complete and correct, in which all cited papers were searched on Pubmed and have not been retracted.

Reply to Reviewer #2 about “6. Review Comments to the Author”:

We deleted the paragraph in the INTRODUCTION part for the azoospermia since this work is not related to the azoospermia at all, according to the suggestion. The reference for this paragraph is the original reference [18]. The original reference [18] was about the study between male infertility and precision medicine and systems proteomics, so it was adjusted to the paragraph: “Male infertility is mainly manifested as azoospermia” in the INTRODUCTION part (line 75-76). For this reason, we adjusted the original reference [18] to reference [7], and references [7-17] to reference [8-18].

Thank you for your help and support!

---

## [Editor Report · Decision Letter 2]

26 Nov 2024

The effects of L-carnitine and fructose in improved Ham's F10 on sperm culture in idiopathic severe asthenospermia within 24h

PONE-D-24-23439R2

Dear Dr. Zhang,

We’re pleased to inform you that your manuscript has been judged scientifically suitable for publication and will be formally accepted for publication once it meets all outstanding technical requirements.

Kind regards,

Xuejiang Guo, Ph.D.

Academic Editor

PLOS ONE

---

## [Editor Report · Acceptance letter]

5 Dec 2024

PONE-D-24-23439R2 

PLOS ONE

Dear Dr. Zhang, 

I'm pleased to inform you that your manuscript has been deemed suitable for publication in PLOS ONE. Congratulations! Your manuscript is now being handed over to our production team.

Kind regards, 

on behalf of

Prof. Xuejiang Guo 

Academic Editor

PLOS ONE